# Robust Adversarial Example Detection Algorithm Based on High-Level Feature Differences

**DOI:** 10.3390/s25061770

**Published:** 2025-03-12

**Authors:** Hua Mu, Chenggang Li, Anjie Peng, Yangyang Wang, Zhenyu Liang

**Affiliations:** 1College of Electronic Engineering, National University of Defense Technology, Hefei 230037, China; wangyy@nudt.edu.cn (Y.W.); liangzy21@nudt.edu.cn (Z.L.); 2The First People’s Hospital of Guangyuan, Guangyuan 628017, China; fightlee8@gmail.com; 3School of Computer Science and Technology, Southwest University of Science and Technology, Mianyang 621010, China; 4Jianghuai Advanced Technology Center, Jianghuai 230031, China; penganjie@swust.edu.cn

**Keywords:** adversarial example detection, feature differences, feature encoder, similarity measurement model, robustness

## Abstract

The threat posed by adversarial examples (AEs) to deep learning applications has garnered significant attention from the academic community. In response, various defense strategies have been proposed, including adversarial example detection. A range of detection algorithms has been developed to differentiate between benign samples and adversarial examples. However, the detection accuracy of these algorithms is significantly influenced by the characteristics of the adversarial attacks, such as attack type and intensity. Furthermore, the impact of image preprocessing on detection robustness—a common step before adversarial example generation—has been largely overlooked in prior research. To address these challenges, this paper introduces a novel adversarial example detection algorithm based on high-level feature differences (HFDs), which is specifically designed to improve robustness against both attacks and preprocessing operations. For each test image, a counterpart image with the same predicted label is randomly selected from the training dataset. The high-level features of both images are extracted using an encoder and compared through a similarity measurement model. If the feature similarity is low, the test image is classified as an adversarial example. The proposed method was evaluated for detection accuracy against four comparison methods, showing significant improvements over FS, DF, and MD, with a performance comparable to ESRM. Therefore, the subsequent robustness experiments focused exclusively on ESRM. Our results demonstrate that the proposed method exhibits superior robustness against preprocessing operations, such as downsampling and common corruptions, applied by attackers before generating adversarial examples. It is also applicable to various target models. By exploiting semantic conflicts in high-level features between clean and adversarial examples with the same predicted label, the method achieves high detection accuracy across diverse attack types while maintaining resilience to preprocessing, providing a valuable new perspective in the design of adversarial example detection algorithms.

## 1. Introduction

The opacity and vulnerability of deep learning models present significant risks, leading to various malicious attacks targeting these applications. The primary threats include model extraction, data poisoning, and adversarial example attacks. Adversarial examples [1,2], first introduced by Szegedy et al., involve the addition of carefully crafted, imperceptible perturbations to original samples, causing models to produce incorrect outputs. Notably, these adversarial examples can often transfer across different models, a phenomenon known as cross-model transferability. This issue arises not only in straightforward tasks, like image classification, but also in more complex applications, such as image segmentation [3] and autonomous driving [4], thus posing substantial challenges to the deployment of deep learning systems.

In response to these threats, a range of defense techniques has emerged. These include robust model architectures designed before the training phase, adversarial training during the training phase, and adversarial example purification and detection during inference [5]. While adversarial training enhances model robustness, it often compromises performance on clean images and may leave models vulnerable to subsequent attacks [6]. Adversarial example purification aims to eliminate adversarial noise but becomes less effective against stronger attacks [7]. Conversely, adversarial example detection offers a flexible solution that operates without modifying the original network architecture, ensuring both efficiency and time-saving benefits [8].

Adversarial example detection algorithms are broadly categorized into supervised and unsupervised methods, depending on their use of adversarial data [9]. Detection techniques include statistical differences, feature differences, input transformations, and reconstruction-based approaches.

**Statistical-based detection:** These methods often use supervised learning, as in Liu et al.’s steganalysis-based algorithm ESRM [10], which employs steganographic features and a binary classifier to detect adversarial perturbations. However, its detection accuracy declines significantly with lower-intensity attacks or optimization-based attacks.

**Feature-based detection:** By analyzing feature differences across network layers, both supervised [11,12] and unsupervised [13] approaches can classify adversarial examples. These methods [11,12] train binary classifiers using various metrics, like local intrinsic dimensionality (LID) and Mahalanobis distance (MD). However, LID estimation may introduce bias in sparse or high-dimensional spaces, while Mahalanobis distance is computationally intensive due to its dependence on training set statistics. In contrast, this method [13] clusters intermediate layer features within the network, allowing for more efficient unsupervised detection of diverse adversarial types.

**Input transformation-based detection:** This category identifies adversarial examples by applying transformations (e.g., denoising, compression, smoothing) and comparing model outputs for original and transformed inputs. These methods, often unsupervised, exploit adversarial sensitivity to such transformations [8,14]. For instance, the FS method [8] uses threshold-based $L_{1}$ distances between predictions pre- and post-transformation and is effective for lower-intensity attacks but less so for stronger ones. The DF method [14] applies scalar quantization and smoothing filters, adjusting parameters with image entropy, although its limited filter capacity can reduce the detection accuracy against high-intensity attacks.

**Reconstruction-based detection:** These methods typically rely on autoencoders or generative models to detect adversarial examples via reconstruction error analysis. For instance, CMAG [15] outperforms Fence FGAN [16] and UADD-GAN [17] by effectively identifying high-quality adversarial samples, though its basic autoencoder may struggle with complex data. ContraNet [18] leverages semantic inconsistencies between inputs and reconstructed images. Its architecture includes an encoder, conditional generator, and similarity measurement model to detect adversarial examples by comparing original images with generated reconstructions.

The field of adversarial example detection encompasses a variety of algorithms, stemming from the absence of a unified theoretical framework governing the generation mechanisms of adversarial examples. Approaches, such as ESRM, FS, and DF, concentrate on the pixel space of images, interpreting adversarial perturbations as forms of steganography or unique noise patterns. Consequently, these methods devise detection algorithms rooted in the analysis of the image space. In contrast, MD shifts its focus to the features extracted by deep neural networks, exploiting the discrepancies between clean images and adversarial counterparts within the feature space to identify adversarial examples. Meanwhile, ContraNet adopts a distinct strategy by leveraging the dissimilarity between an adversarial image and a clean image, where the clean image is generated by projecting the discriminative features of the adversarial image extracted by the DNN back into the input space. This diversity in methodological approaches underscores the complexity and multifaceted nature of adversarial example detection in the realm of deep learning.

While these detection methods have demonstrated promising results, their performance is critically constrained by the reliance on either meticulous parameter tuning or sophisticated feature selection, significantly limiting their scalability and practicality in real-world applications. Additionally, the robustness of detection algorithms against preprocessing operations has not received adequate attention, highlighting a critical gap in the current research. The primary objective of this study is to develop an adversarial example detection method that achieves high detection accuracy, exhibits strong robustness to diverse input corruptions, and demonstrates suitability for target models while eliminating the need for extensive manual optimization or complex feature engineering.

Our proposed method is inspired by ContraNet, but it differs in key aspects. Specifically, we leverage high-level feature differences in the examples instead of projecting the high-level feature back into images as in ContraNet. We compare a test example with a clean example selected from the training set with the same predicted label as the test sample, rather than comparing it with a reconstructed sample. This approach eliminates the need for a conditional generative network to reconstruct images, resulting in enhanced computational efficiency and real-time performance. To the best of our knowledge, this represents the first work in the field of adversarial example detection to leverage high-level features from image pairs sharing the same predicted label. In summary, the following contributions are presented:This study demonstrates that high-level feature distinctions are sufficient for detecting adversarial examples, providing a novel perspective on AE detection.This study proposes a new detection method utilizing a similarity measurement model trained on two types of concatenated high-level feature data, one is from an adversarial example and a clean example sharing the same predicted label, and the other from two clean examples with the same predicted label.This study validates the detector’s performance across various attack algorithms and neural network architectures, confirming its robustness against different preprocessing operations.

The remaining of this paper is organized as follows. Section 2 presents a systematic review of three key components integral to our experimental framework: (1) state-of-the-art attacking algorithms, (2) contemporary adversarial example detection methodologies, and (3) prevalent image corruption techniques applied prior to adversarial example generation. Section 3 presents our novel detection approach based on high-level feature differences (HFDs). Then, extensive experiments are conducted in Section 4 to systematically evaluate the performance of our method against different attack types, attack intensities, and preprocessing operations prior to adversarial example generation. Furthermore, the algorithm’s applicability to different target models has also been thoroughly examined. Finally, Section 5 concludes our research findings and outlines potential directions for future work in this domain.

## 2. Related Work

This section provides a brief overview of the algorithms for generating adversarial examples, existing detection algorithms, and image corruptions that may influence the accuracy of these detection methods.

### 2.1. Attacking Algorithms

Szegedy et al. [1] were the first to observe that deep neural network (DNN) models are vulnerable to adversarial perturbations. An adversarial example is an input crafted by an adversary to produce an incorrect output from a target machine learning model. Let *x* be a legitimate image, and *y* be the corresponding class label. For a well-trained DNN model f, f(x)=y. An adversarial attack aims to find an adversarial $x_{adv}$ for *x* such that $f(x_{adv})\neq y$. Several techniques have been proposed to find adversarial examples. Below, we present the adversarial attack algorithms employed in our experiments, which encompass a range of strengths and complexities, as well as both white-box and black-box attack scenarios. Specifically, BIM, MI-FGSM, and SINI-FGSM are primarily white-box attacks, with increasing strength and complexity. Among these, MI-FGSM and SINI-FGSM have good transferability, making them effective in black-box settings. AutoAttack is explicitly designed to address both white-box and black-box scenarios, establishing itself as a robust benchmark for evaluating adversarial robustness.

Fast Gradient Sign Method: FGSM

Goodfellow et al. [2] proposed the fast gradient sign method (FGSM) to efficiently generate adversarial examples. FGSM finds an adversarial example $x_{adv}$ as follows:(1)$$x_{adv}=x+\varepsilon \cdot sign(\nabla_{x}L(f(x),\;y))$$
where ε is a constant controlling the maximum perturbation per pixel, sign denotes the symbolic function, L(⋅) denotes the loss function, and $\nabla_{x}$ represents the gradient of the loss function with respect to the input x.

2.Basic Iterative Method: BIM

Kurakin et al. [19] extended FGSM to an iterative algorithm, named the basic iterative method (BIM), which applies multiple small-step updates rather than a single-step update along the gradient direction.

3.Momentum Iterative Fast Gradient Sign Method: MI-FGSM

Dong et al. [20] introduced momentum-based iterative algorithms to improve the transferability of adversarial examples. The momentum iterative fast gradient sign method (MI-FGSM) incorporates a momentum term in each iteration to avoid poor local maxima.

4.Scale-Invariant Nesterov Iterative Fast Gradient Sign Method: SINI-FGSM

Lin et al. [21] proposed SINI-FGSM, which enhances the traditional iterative FGSM by combining a Nesterov accelerated gradient (NAG) and scale invariance. The NAG improves convergence speed, while scale invariance addresses sensitivity to input scaling, enhancing robustness against defenses that rely on gradient masking or input transformations.

5.AutoAttack: AA

AutoAttack [22] is an ensemble of parameter-free adversarial attacks designed to reliably assess model robustness. It combines several attacks, including white-box and black-box methods, providing a comprehensive evaluation tool.

### 2.2. Methods of Adversarial Example Detection Detecting Adversarial Examples

We discuss the adversarial example detection algorithms compared with our proposed algorithm in our experiments. These include two supervised algorithms, i.e., the Mahalanobis distance method and enhanced spatial rich model, and two unsupervised algorithms, i.e., feature squeezing and detection filter.

Feature Squeezing method: FS

Xu et al. [8] proposed a strategy utilizing feature squeezing to detect adversarial examples. The detector considers an input adversarial if the difference between predictions on the original and squeezed samples exceeds a threshold. Two feature-squeezing methods, namely reducing color bit depth and spatial smoothing, allow effective detection of static adversarial examples.

2.Mahalanobis Distance method: MD

Lee et al. [12] proposed a method applicable to any pretrained softmax neural classifier, detecting abnormal test samples using a distance-based approach. Under the assumption that class-conditional distributions are Gaussian, the Mahalanobis distance is used to calculate confidence scores, and a two-class classifier is then trained to distinguish clean and adversarial examples.

3.Enhanced Spatial Rich Model: ESRM

Liu et al. [10] proposed a detection algorithm from the steganalysis point of view. Both adversarial attacks and steganography on images cause perturbations on the pixel values. By modeling the differences between adjacent pixels in natural images, it is possible to identify deviations due to adversarial attacks. The spatial rich model (SRM) with 34,671 features is a kind of steganalysis feature set recommended to detect adversarial examples. A detector was trained with enhanced SRM features to detect adversarial examples.

4.Detection Filter: DF

Liang et al. [14] treated perturbations as noise and applied scalar quantization and smoothing filters to mitigate their effects. An input is classified as adversarial if the label of the original input differs from the filtered version, without requiring prior knowledge of attack types.

Although the aforementioned adversarial example detection algorithms demonstrate considerable efficacy, it is imperative to underscore that their implementation and optimization constitute a non-trivial endeavor, as their performance critically relies on either meticulous parameter tuning or sophisticated feature selection. Specifically, FS necessitates precise selection of squeezing thresholds, MD demonstrates significant sensitivity to the feature space configuration for distance computation, ESRM demands rigorous feature engineering and feature selection, and DF requires dataset-specific parameter optimization to ensure robust performance.

### 2.3. Downsampling and Common Corruptions

Before generating adversarial examples, images are typically subjected to preprocessing steps, such as downsampling. Additionally, certain factors, including imaging conditions, image compression, and attacker-driven preprocessing operations, impact image quality. Collectively, these effects are referred to as “common corruptions”. Both downsampling and common corruptions may significantly influence the detection performance of adversarial example detectors. However, the impact of these factors has received limited attention in existing research.

Downsampling is a frequently applied operation in adversarial image generation. To conform to the smaller input dimensions required by neural networks, attackers often downsample images prior to launching an attack. Various adversarial attack frameworks incorporate specific downsampling algorithms, such as the bilinear interpolation utilized by Cleverhans [23] and the nearest-neighbor interpolation employed by EvadeML [8].

In this work, we select six types of corruption from the IMAGENET-C benchmark dataset [24], which is widely used to evaluate the corruption robustness of image classifiers. Gaussian noise, for example, commonly arises in low-lighting conditions, while zoom blur occurs when a camera moves rapidly toward an object. Frost artifacts can form when ice crystals accumulate on lenses or glass surfaces, impairing image clarity. Contrast changes, affected by lighting conditions and object color, may result in either high or low visibility. Elastic transformations deform small regions within an image through localized stretching or contraction. Lastly, JPEG compression, a lossy format, introduces visual artifacts that could influence the detection accuracy of adversarial examples.

## 3. HFD: Proposed Method

In this section, we detail the proposed HFD design. First, we introduce the motivation behind our design. Then, we depict an overview of HFD’s implementation. Next, we discuss the core component, i.e., the similarity measurement model, as well as its training process. Finally, we provide the detecting process for HFD.

### 3.1. Design Motivation

Following the common consensus that DNNs are feature extractors or encoders, we analyze adversarial examples from the feature perspective. Adversarial images are generated by adding small-norm perturbations (e.g., $L_{\infty }$ or $L_{2}$) that are imperceptible or negligible to humans. While these perturbations are minimal at the pixel level, they amplify as the image propagates through convolutional networks [25], hallucinating non-existent activations in feature maps. This amplification overwhelms true signal activations, causing incorrect predictions [26]. At the same time, revealed that image-dependent adversarial examples include not only the features of the target class, but also the original image features [27].

In our work, high-level features are utilized to detect adversarial examples. These features refer to the activations of the final convolutional layer, located just before the fully connected layer. Our design is motivated by the insight that high-level features of an adversarial example and a clean image sharing the same predicted class label exhibit lower similarity compared to a pair of clean images belonging to the same class, as illustrated by Figure 1. Specifically, for a clean image, the high-level features predominantly reflect the ground-truth class characteristics. In contrast, the high-level features of an adversarial example not only incorporate the target class features but also retain residual features from the original class. Consequently, the high-level features of two clean images belonging to the same class are more similar than that of an image pair composed of an adversarial example and a clean example sharing the same predicted label. This distinction enables the training of a binary classifier to differentiate between these two types of image pairs. Since this binary classifier operates based on the similarity of feature pairs, it is termed a similarity measurement model. Such a model could, thus, be utilized for adversarial example detection.

We opt for high-level features over middle-level features due to the progressive amplification of perturbations as an adversarial image propagates through the network [1]. Specifically, the similarity between the features of an adversarial example and a clean image with the same predicted label increases with deeper layers, making it increasingly challenging to differentiate between the two types of image pairs at higher levels. Therefore, a similarity measurement model must be capable of distinguishing high-level features to effectively detect adversarial examples.

### 3.2. Implementation Overview

It can be inferred from Figure 1 that HFD should consist of two components, namely an encoder (*E*) and a similarity measurement model, which is a discriminator (*D*) to judge an input image x is adversarial or not. To be more specific, the encoder E is the target model f excluding the last fully connected layer and output layer and is in charge of extracting the high-level features of the input image. The similarity measurement model *D* measures the similarity of high-level features between the input image x and its counterpart x′, which is randomly picked up from the training set $X_{train}$ with the same predicted label as x.

During inference, given an input $x\in X_{test}$, we first obtain the predicted label y  from the target model f and then pick up an image $x\prime \in X_{train}$ such that $f\left(x\prime \right)=y$. In the following process, the encoder *E* will retrieve high-level features for the images x and x′, respectively. Lastly, the similarity measurement model will evaluate the similarity between E(x) and E(x′), to identify AEs. Specifically, for a legitimate input that is correctly inferred, E(x) and E(x′) are similar. Otherwise, for an AE, the high-level features would have low similarity.

### 3.3. The Similarity Measurement Model

The similarity measurement model D is essentially a binary classifier whose input is the concentrated features of an input image and its counterpart. The output of model D is 1 or 0, representing whether the input image is adversarial or not, respectively. We designed the similarity measurement model in the framework of an MLP (multilayer perception). There are 2 hidden layers, with 3000 and 500 neurons, respectively. The output layer has two neurons. The activation function for all the neurons is a sigmoid function.

The training procedure for the similarity measurement model is depicted in Algorithm 1 and illustrated by Figure 2.

**Algorithm 1:** Training the similarity measurement model.**Input**: Xtrain, selection operation S(·) , training epoch T, target model f**Output**: the parameters of model D1: for *t* = 0 to *T*-1 do2:       feed x∈Xtrain into the target model f and obtain the label y=f(x)3:       select a counterpart image for x , x′=Sy,  where  x′∈Xtrain4:       generate an adversarial example xadv by attacking x5:       feed xadv into f and obtain yadv=f(xadv)6:       select a counterpart image for xadv , xadv′=Syadv , where xadv′∈Xtrain7:       feed x, xadv, x′, xadv′ into the Encoder E:           Fx=Ex,Fxadv=Exadv,Fx′=Ex′,Fx′adv=Ex′adv8:       construct feature pairs:           Fsim=concatFx,Fx′, labeled as 0;          Fdif=concat(Fxadv,Fxadv′), labeled as 19:       feed Fsim and Fdif to model D10:      optimize model D using cross-entropy loss11: end for

In this algorithm, the similarity measurement model is trained to measure semantic similarity based on feature pairs of an image and its counterpart. For each benign instance $x\in X_{train}$, the target model f provides its predicted label y=f(x). A counterpart image $x^{\prime }$ is selected from $X_{train}$ by the selection operation *S*(y), based on the criterion of sharing the same predicted label y; this counterpart is a benign instance and exhibits similar semantics with x. An adversarial instance $x_{adv}$ is then generated by perturbing x. For $x_{adv}$, a benign counterpart $x_{adv}^{\prime }$ is also selected by operation *S*$\left(y_{adv}\right)$, which shares the same label $y_{adv}\;$ but has divergent semantics from $x_{adv}$.

The encoder, which is the target model f excluding the last fully connected layer and output layer, extracts high-level features from each instance. Two types of feature pairs are created: (1) a similarity pair (two benign samples belonging to the same class, labeled as 0, top panel of Figure 2); and (2) a difference pair (one benign sample and one adversarial sample with the same predicted label, labeled as 1, bottom panel of Figure 2). These pairs are fed into the similarity measurement model *D*, which is optimized using cross-entropy loss. A dropout strategy is adopted for the two hidden layers during training. This training approach enables the model to effectively measure semantic similarity for a pair of images with the same predicted labels.

### 3.4. Detecting Adversarial Examples

The inference procedure is straightforward. Algorithm 2 describes the detecting process of the detector, which is composed of the similarity measurement model *D* and the encoder *E*. The process begins by passing the test image x through the target model f to obtain the predicted label y. A counterpart benign image $x^{\prime }$ is selected from the training dataset $X_{train}$ using the selection operation $S\left(y\right)$, which ensures that $x^{\prime }$ shares the same predicted label as x. High-level feature representations for both images x and $x^{\prime }$ are extracted by the encoder *E* and then concatenated to form a combined feature vector $F_{in}$. This concatenated feature vector is then fed into the similarity measurement model *D* to produce a binary output, where 1 indicates an adversarial instance and 0 indicates a benign instance.

**Algorithm 2:** Detecting adversarial examples.**Input:** image x, detector **Output:**  output=1, image x is adversarial; output=0, image x is benign1: y=f(x)2: x′=Sy3: Fx=Ex,Fx′=Ex′4: Fin=concatFx,Fx′5: result=D (Fin)6:     if result = 17:       image x is adversarial8:     else9:       image x is benign

## 4. Verification

### 4.1. Experimental Settings

**Attacks**. Four popular attacks, namely BIM, MI-FGSM, SINI-FGSM, and AutoAttack, are employed to craft adversarial examples to evaluate the proposed detector. The magnitude of adversarial perturbations ϵ is set as 4, 8, and 16, while the other hyperparameters are consistent with the settings in the reference papers.

**Datasets**. We conduct experiments on subsets of ILSVRC 2012 [28]. The training dataset for the similarity measurement model D is composed of 10,000 images, with 10 randomly selected from each class of the ILSVRC 2012 training set. The clean images to test are chosen from the ILSVRC 2012 test set, with 10 for each class and 10,000 images altogether. Adversarial examples are generated from correctly classified clean images, resulting in 10,000 image pairs for testing the detector.

**Target models**. For target models, we consider classical convolutional neural networks, including ResNet-50, Inception V3, EfficientNet B0, and ConvNeXt. ResNet-50 [29] is widely adopted due to its balance of depth and efficiency, achieving strong performance across various computer vision tasks. ConvNeXt [30] is a recently proposed advanced convolutional neural network classification model built upon ResNet, incorporating techniques from vision transformers. Inception V3 [31] employs multi-scale feature extraction with parallel convolutional filters. Although it is more computationally intensive, it achieves higher accuracy than ResNet-50. EfficientNet B0 [32] is designed for high accuracy with minimal resources and is highly efficient for mobile and edge devices, featuring only 18 convolutional layers.

**Threat model**. The defender has access only to information about the target model, with no knowledge of the attacker’s strategy. The adversarial examples used for training the detector are generated exclusively using the BIM algorithm, with the perturbation strength set to ϵ = 4. Retraining of the detector is required only when the target model undergoes changes.

**Evaluation metric**. The metric should evaluate a detector from two perspectives: (1) the impact on classification accuracy on legitimate samples, and (2) the ability of AE detection. We conventionally denote the adversarial example as the Positive sample and the clean sample as the Negative sample. We define *acc* as the evaluation metric for the detector. Equation (2) is as follows:(2)$$acc=\frac{N_{p}+P_{r}}{N+P}$$
where N is the number of the negative samples, P is the number of the positive samples, $N_{p}$ denotes the number of negative samples that passed the detector, and $P_{r}$ represents the number of positive samples which are rejected by the detector. It is reasonably straightforward to deduce that the *acc* metric serves as an integral indicator, encapsulating both the true positive rate (TPR) and the false positive rate (FPR). Furthermore, in our experimental setup, we employ the configuration of N=P. Consequently, a high accuracy value inherently implies an elevated TPR coupled with a diminished FPR.

**Baselines.** We choose four high cited detection-based defense methods as the baselines, including the ESRM (enhanced spatial rich model), FS (feature squeezing), MD (Mahalanobis distance method) and DF (detection filter) approaches. Thanks to their authors, we reproduce their work with officially open-source codes. FS combines three types of squeezers, where the threshold T is set for FPR = 5%. For ESRM and MD, 5000 image pairs randomly picked from the test dataset are used to train the binary classifiers, while the remaining pairs are used to evaluate the detectors.

### 4.2. Detection Accuracy

We evaluate the performance of our method and the baseline detectors on subsets of ILSVRC 2012 with the ResNet-50 classifier. The detector is trained with BIM adversarial examples where ϵ = 4, and it is then used to detect adversarial examples generated from four typical attacks with different magnitudes of disturbance, including BIM, MI-FGSM, SINI-FGSM, and AutoAttack. The experimental settings are listed in Table 1.

The results are shown in Table 2 and Figure 3. It can be seen that the proposed method significantly outperforms the FS, DF, and MD detection algorithms in detection accuracy and achieves comparable accuracy to the ESRM detection algorithm on the target model ResNet-50. Notably, as the intensity of adversarial attacks increases, the detection accuracy of the FS, DF, and MD algorithms shows a clear downward trend, with the maximum drop reaching 27.17%. In contrast, the detection accuracy of the proposed method remains relatively stable, indicating strong robustness against varying attack intensities. Furthermore, as can be observed from Table 2 and Figure 3, although the detector is trained with BIM adversarial examples with ϵ = 4, the detection accuracy of the proposed method remains relatively stable across different attack types and different magnitudes of disturbance, demonstrating good robustness against attacks.

Since only ESRM among the four baseline methods achieves comparable accuracy to the proposed method, ESRM continues to serve as the baseline in the following robustness experiments.

### 4.3. Robustness to Downsampling

To analyze the impact of downsampling on detection accuracy, we take the detection of BIM attacks as an example and select ESRM as the baseline for comparison. The experimental settings are listed in Table 3.

The primary factor in downsampling algorithms is the interpolation kernel. In the experiments, we used the nearest, bilinear, and Lanczos interpolation kernels in the Resize function in PyTorch 2.6.0 for downsampling. The nearest interpolation kernel selects the value of the nearest original image pixel as the value of the target pixel after downsampling. This method is simple and fast to implement, but it can result in jagged edges or distortions in the image. The bilinear interpolation kernel calculates the pixel value after downsampling by taking a weighted average of the four nearest pixels surrounding the target pixel. Compared to the nearest interpolation kernel, the bilinear interpolation kernel can reduce jagged edges and distortions. The Lanczos interpolation kernel is based on a sine function and takes more pixels into account when calculating new pixel values, producing a smoother image.

The experimental results in Table 4 indicate that the proposed method’s detection accuracy remains stable across different downsampling techniques. Specifically, the maximum fluctuation in detection accuracy across the nearest, bilinear, and Lanczos downsampling methods is 0.31%, 0.13%, and 0.13% for ϵ = 4, ϵ = 8, and ϵ = 16, respectively. In contrast, the ESRM detection algorithm, which is based on steganalysis, is more sensitive to downsampling, with a variation in detection accuracy exceeding 2%.

### 4.4. Robustness to Common Corruptions

To evaluate the robustness of the proposed method against image preprocessing before generating adversarial examples, we conducted experiments to measure detection accuracy under various preprocessing operations. Assuming that the defender is unaware of the preprocessing operations, the detector was trained solely with clean samples and adversarial examples without any preprocessing. The target model used was ResNet-50, with the BIM attack method. Consistent with the downsampling experiment, ESRM was selected as the baseline for comparison. The experimental settings in Table 3 are adopted for this experiment.

Considering that the adversarial examples are required to be imperceptible to human eyes, we select the preprocessing operations that have a minimal impact on image quality, including Gaussian noise, zoom blur, frost, contrast enhancement, elastic transform, and JPEG compression. The parameters for these operations are set to ensure the corruption’s imperceptibility to human eyes. Specifically, the mean of the Gaussian noise is set to 0, and the standard deviation is set to 0.02. The JPEG compression factor is set to 95. The other parameter settings follow the work of Hendrycks et al. [24].

The experimental results are presented in Table 5. It can be observed that the proposed HFD method exhibits strong robustness against six different types of corruptions, while the performance of ESRM is comparatively weaker. Specifically, for adversarial examples subjected to these six corruptions, the detection accuracy of HFD deviates by no more than 1% compared to that of uncorrupted adversarial examples. In contrast, the detection accuracy of ESRM drops by over 47% when dealing with adversarial examples preprocessed with Gaussian noise, and it also experiences a decline of more than 1% for adversarial examples corrupted with Frost and JPEG preprocessing.

### 4.5. Suitability for Target Models

We selected three neural network structures in addition to ResNet-50 to verify the suitability for different target model structures, including Inception V3, EfficientNet B0, and ConvNeXt. Specifically, for each target model, the encoder needs to be updated accordingly, and the similarity measurement model needs to be retrained. The experimental settings for the target model, Inception V3, are detailed in Table 6. For EfficientNet B0 and ConvNeXt, the experimental setup remains consistent with that of Table 6, with the exception being that the models used in both the design phase and the detection phase are replaced by EfficientNet B0 and ConvNeXt, respectively.

The experimental results for the four target models are presented in Table 7. Among them, the results for ResNet50 are repeated from Table 2 to ensure a comprehensive presentation of all results. From Table 7, it is evident that the proposed adversarial example detection method achieves high detection accuracy across different target models, demonstrating its broad applicability. Among these, ConvNeXt has the highest detection accuracy, likely due to its encoder’s superior feature extraction capability, which improves the similarity measurement model’s ability to distinguish between clean and adversarial samples. The proposed method is also effective against various attack algorithms, with AutoAttack being the most difficult to detect across different target models.

## 5. Discussions

The effectiveness of HFD is somewhat surprising, given its simplicity and low computational cost compared to other detection algorithms. Although developing a comprehensive theory of adversarial examples detection remains an elusive goal, the success of HFD provides a valuable insight: even subtle differences in high-level feature representations between adversarial examples and clean images—despite their shared predicted label—can be leveraged for effective detection. This finding highlights the potential of exploiting minor feature discrepancies for robust detection, regardless of the adversarial example generation mechanism.

Although our experiments have thus far been confined to image classification models, the HFD approach, which leverages feature representations rather than raw image data, holds promise for broader applicability across various domains where deep learning is employed. For instance, in speech recognition systems, HFD could potentially be adapted to defend against adversarial voice commands by analyzing the semantic features of audio data. This versatility underscores the potential of HFD as a generalizable defense mechanism against adversarial attacks in diverse deep learning applications, contributing to the development of more reliable and trustworthy AI systems.

The HFD method, while effective, has several potential limitations. First, because the encoder shares the same structure as the classifier, the detector must be retrained for different classifier architectures, which could limit its practicality in diverse deployment scenarios. Second, the method relies on differences in feature representations, making it potentially less effective against adversarial examples generated with low-magnitude perturbations (i.e., weak attack strength). Finally, the detector itself is implemented as a neural network, introducing a vulnerability: if the detection strategy is exposed to attackers, the detector could become a target of adversarial attacks, compromising its effectiveness. These limitations highlight areas for future research, such as improving robustness against weak adversarial attacks, developing architecture-agnostic detectors, and enhancing the security of the detector itself against adversarial targeting.

## 6. Conclusions

To strengthen the defense capability of CNN classification networks against adversarial examples, this study presents a novel detection method that leverages semantic conflicts in high-level features between adversarial and clean examples sharing the same predicted label. Importantly, our approach does not require the adversarial example to be generated from the corresponding clean example; instead, the only prerequisite is that both examples yield the same prediction.

In detection accuracy experiments, 12 attack scenarios were designed by applying 4 attack algorithms (BIM, MI-FGSM, AutoAttack, and SINI-FGSM) to the ResNet50 target model, each with 3 intensity levels. The results (Table 2 and Figure 2) show that the proposed method significantly outperforms FS, DF, and MD, while achieving accuracy comparable to ESRM.

Robustness to downsampling and corruptions tests were performed under three BIM attack intensities, comparing the proposed method with ESRM. The proposed method showed a maximum accuracy fluctuation of 0.31%, far lower than ESRM’s 2.18–3.57% across three downsampling operations (Table 4). Against six common corruptions, the proposed method remained stable with a maximum deviation of 0.87%, while ESRM’s accuracy dropped by over 47% under Gaussian noise.

To evaluate generalizability, the method was tested on three additional target models (Inception V3, EfficientNet B0, and ConvNeXt) under four attack algorithms with three intensity levels each. Across these 48 scenarios, the proposed method consistently achieved a detection accuracy above 89% (Table 7), demonstrating its applicability to diverse target models.

In future work, we plan to explore scenarios in which the HFD method is exposed to adaptive attacks, where adversarial examples are crafted not only to deceive the target model but also to bypass the detection mechanism. This challenge could be addressed by retraining the similarity measurement model with updated adversarial examples, enhancing its robustness against such adaptive strategies.

## Figures and Tables

**Figure 1 sensors-25-01770-f001:**
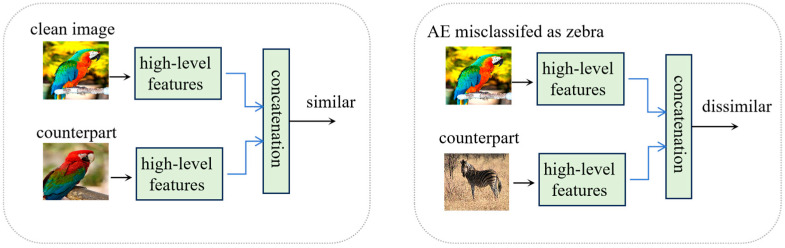
Two types of high-level features of image pairs: (**left**) a pair of clean images belonging to the same class ‘parrot’; (**right**) an adversarial example and a clean image sharing the same predicted class label ‘zebra’.

**Figure 2 sensors-25-01770-f002:**
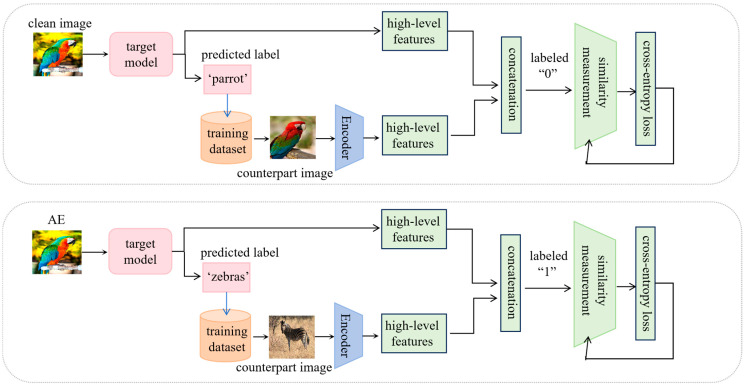
Training of the similarity measurement model. The top panel shows the training sample of a similarity pair, and the bottom panel shows a difference pair.

**Figure 3 sensors-25-01770-f003:**
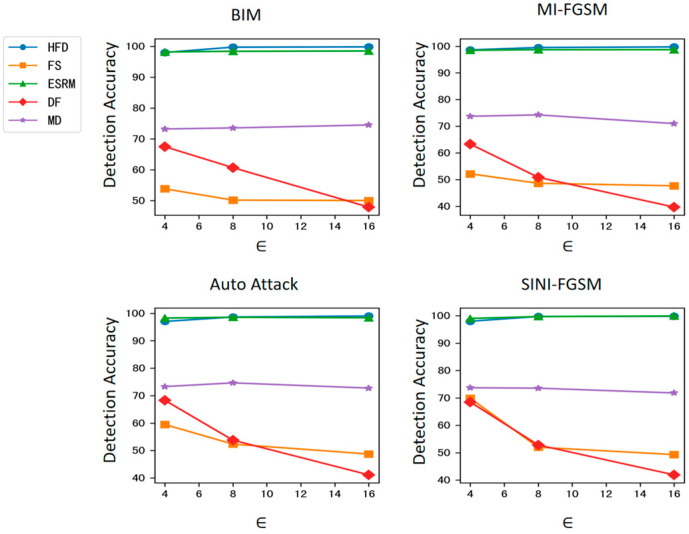
Detection accuracy against attacks with different magnitudes of disturbance on ResNet-50.

**Table 1 sensors-25-01770-t001:** Experimental settings for the detection accuracy experiment.

	Design Phase	Detection Phase
Model	ResNet-50	ResNet-50
Dataset	Subsets of ILSVRC 2012	Subsets of ILSVRC 2012
Attack	BIM(ϵ = 4)	BIM(ϵ = 4,8,16) MI-FGSM(ϵ = 4,8,16) SINI-FGSM(ϵ = 4,8,16) AutoAttack(ϵ = 4,8,16)

**Table 2 sensors-25-01770-t002:** Comparison of detection accuracy against attacks with different magnitudes of disturbance on ResNet-50.

Attack Method	Detection Method	ϵ = 4	ϵ = 8	ϵ = 16
BIM	HFD	97.98 *	**99.72**	**99.80**
	FS	53.81	50.15	50.02
	ESRM	**98.17**	98.37	98.48
	DF	67.47	60.68	47.89
	MD	73.20	73.55	74.50
MI-FGSM	HFD	**98.60**	**99.50**	**99.74**
	FS	52.17	48.62	47.67
	ESRM	98.47	98.69	98.72
	DF	63.33	50.86	39.69
	MD	73.74	74.25	71.00
AutoAttack	HFD	97.04	**98.65**	**99.02**
	FS	59.48	52.33	48.72
	ESRM	**98.26**	98.56	98.36
	DF	68.30	53.78	41.13
	MD	73.30	74.65	72.75
SINI-FGSM	HFD	97.99	**99.72**	99.80
	FS	69.84	51.99	49.30
	ESRM	**99.01**	99.69	**99.90**
	DF	68.50	52.80	41.90
	MD	73.70	73.55	71.80

* For adversarial examples generated by the same attack algorithm at the same attack strength. The highest detection accuracy across different methods is shown in bold.

**Table 3 sensors-25-01770-t003:** Experimental settings for downsampling robustness.

	Design Phase	Detection Phase
Model	ResNet-50	ResNet-50
Dataset	Subsets of ILSVRC 2012	Subsets of ILSVRC 2012
Attack	BIM(ϵ = 4)	BIM(ϵ = 4,8,16)

**Table 4 sensors-25-01770-t004:** Comparison of detection accuracy (%) between HFD and ESRM under different downsampling operations.

Downsampling	HFD			ESRM		
	ϵ = 4	ϵ = 8	ϵ = 16	ϵ = 4	ϵ = 8	ϵ = 16
Nearest	97.92	99.59	99.67	95.84	96.22	95.01
Bilinear	97.98	99.72	99.80	98.68	98.22	98.54
Lanczos	97.67	99.66	99.70	98.33	98.40	98.58
**Fluctuation**	0.31	0.13	0.13	2.84	2.18	3.57

**Table 5 sensors-25-01770-t005:** Comparison of detection accuracy (%) between HFD and ESRM under different common corruptions. The corruptions resulting in maximum deviation are shown in bold.

Corruptions	HFD			ESRM		
	ϵ = 4	ϵ = 8	ϵ = 16	ϵ = 4	ϵ = 8	ϵ = 16
None	97.98	99.72	99.80	98.17	98.37	98.48
Gaussian	**98.85**	99.59	**99.60**	**50.32**	**50.43**	**50.82**
Zoom blur	98.09	99.76	99.81	99.30	99.31	99.35
Frost	98.36	99.68	99.71	96.02	96.49	96. 73
Contrast	98.62	99.80	99.84	98.88	99.18	99.74
Elastic	98.04	**99.87**	99.93	99.78	99.89	99.91
JPEG	98.05	99.85	99.89	96.47	97.29	97.38
**Maximum** **deviation**	0.87	0.15	0.2	47.85	47.94	47.66

**Table 6 sensors-25-01770-t006:** Experimental settings for the target model of Inception V3.

	Design Phase	Detection Phase
Model	Inception V3	Inception V3
Dataset	Subsets of ILSVRC 2012	Subsets of ILSVRC 2012
Attack	BIM(ϵ = 4)	BIM(ϵ = 4,8,16) MI-FGSM(ϵ = 4,8,16) SINI-FGSM(ϵ = 4,8,16) AutoAttack(ϵ = 4,8,16)

**Table 7 sensors-25-01770-t007:** Detection accuracy (%) of HFD for different target models against different attacks.

Model	Attack	ϵ = 4	ϵ = 8	ϵ = 16
ResNet 50	BIM	97.98	99.72	99.80
	MI-FGSM	98.60	99.50	99.74
	AutoAttack	97.04	98.65	99.02
	SINI-FGSM	97.99	99.72	99.80
Inception V3	BIM	96.63	98.46	98.47
	MI-FGSM	98.29	98.72	98.77
	AutoAttack	89.36	92.78	93.67
	SINI-FGSM	93.63	96.47	97.93
EfficientNet B0	BIM	97.31	97.78	97.79
	MI-FGSM	97.76	97.92	98.00
	AutoAttack	93.27	93.98	94.12
	SINI-FGSM	94.72	97.78	97.89
ConvNeXt	BIM	98.86	99.08	99.30
	MI-FGSM	99.15	99.35	99.46
	AutoAttack	98.39	99.03	99.22
	SINI-FGSM	97.39	99.05	99.06

## Data Availability

The datasets applied in this paper are all open-source datasets. The open source URL is shown below. https://image-net.org/challenges/LSVRC/2012/2012-downloads.php, accessed on 31 October 2024.

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
