# Peer review of "Robust Adversarial Example Detection Algorithm Based on High-Level Feature Differences"

_sensors, 2025, doi:10.3390/s25061770_

Round 1

Reviewer 1 Report

Comments and Suggestions for Authors

The paper proposes a novel adversarial example detection algorithm based on HFD to improve the robustness of detection. It has high detection accuracy, outperforming many existing methods, and is robust against different attack types and intensities. The algorithm's applicability to preprocessing operations and different target models has been comprehensively evaluated.

Suggestions for Revision:

At the end of the introduction, use a paragraph to introduce the structure of the article.

In the "2. Related Work" section, the advantages and disadvantages of various attacking algorithms and detection algorithms, as well as their comparisons and correlations, can be further elaborated to better highlight the advantages of the proposed HFD algorithm.

In the experimental part, more experimental settings and comparative experiments can be added, such as different datasets, combinations of different attack parameters, etc., to more comprehensively evaluate the performance of the HFD algorithm.

The theoretical analysis of the algorithm can be further deepened, for example, a more in-depth exploration of the nature and role of high-level feature differences, to strengthen the theoretical foundation of the algorithm.

Reviewer 2 Report

Comments and Suggestions for Authors

The article contains quite interesting results of the (heuristic?) method key part of which is an adversarial example detection algorithm based on High-level Feature Differences. Basic remarks and recommendations:

- the goal of the research should be clearly formulated in terms of improving the relevant characteristics of the algorithms. This should be traced in research materials and conclusions. From the analysis of the article, it follows that it is about improving detection accuracy, robustness and suitability... (subsections 4.2-4.5);

- the scientific novelty of the proposed method must be clearly defined by general theoretical components. The HFD based method is heuristic or rigorously proven. This is not clear from section 3.1 and needs clear clarification and confirmation;

- the authors compare the proposed method (algorithm) with known ones based on several indicators/metrics. Is such a comparison possible for the integral indicator?

- is the number of experiments for evaluating the indicators of the proposed method sufficient? It is necessary to briefly justify;

- what limitations does the proposed method have in its software or software-hardware implementation compared to known algorithms? It must be described;

- the authors do not mention such important characteristics of AI tools as trustworthiness, explainability, ethics. Perhaps it is advisable to analyze them as well;

- the previous comments are one of the arguments to add a special "Discussion" section.

Round 2

Reviewer 2 Report

Comments and Suggestions for Authors

Only minor remarks: captions to the tables 1,3.

Author Response

Comments 1:Only minor remarks: captions to the tables 1,3.

Response 1: Thank you for your feedback. I have revised the table captions as follows:

  1. The original caption for Table 1, "experimental settings for the detection accuracy experiment", has been updated to "Experimental settings for detection accuracy evaluation.".
  2. The original caption for Table 3, "experimental settings for the robustness to downsampling experiment", has been changed to "Experimental settings for downsampling robustness.".

I believe the updated version is more concise and formal. Thank you again for improving the quality of the manuscript. 
